# Neurodevelopment Is Dependent on Maternal Diet: Placenta and Brain Glucose Transporters GLUT1 and GLUT3

**DOI:** 10.3390/nu16142363

**Published:** 2024-07-21

**Authors:** Tomoko Daida, Bo-Chul Shin, Carlos Cepeda, Sherin U. Devaskar

**Affiliations:** 1Department of Pediatrics, Division of Neonatology and Developmental Biology and Neonatal Research Center, at the UCLA Children’s Discovery and Innovation Institute, David Geffen School of Medicine at UCLA, Los Angeles, CA 90095, USA; tdaida@mednet.ucla.edu (T.D.); bshin@mednet.ucla.edu (B.-C.S.); 2Intellectual and Developmental Disabilities Research Center and Brain Research Institute, David Geffen School of Medicine at UCLA, Los Angeles, CA 90095, USA

**Keywords:** glucose transporter, glucose transporter 3, maternal diet, postnatal diet, neurodevelopment, neurodevelopmental disorders, neurodegenerative disorders, Huntington’s disease, brain, placenta

## Abstract

Glucose is the primary energy source for most mammalian cells and its transport is affected by a family of facilitative glucose transporters (GLUTs) encoded by the SLC2 gene. GLUT1 and GLUT3, highly expressed isoforms in the blood–brain barrier and neuronal membranes, respectively, are associated with multiple neurodevelopmental disorders including epilepsy, dyslexia, ADHD, and autism spectrum disorder (ASD). Dietary therapies, such as the ketogenic diet, are widely accepted treatments for patients with the GLUT1 deficiency syndrome, while ameliorating certain symptoms associated with GLUT3 deficiency in animal models. A ketogenic diet, high-fat diet, and calorie/energy restriction during prenatal and postnatal stages can also alter the placental and brain GLUTs expression with long-term consequences on neurobehavior. This review focuses primarily on the role of diet/energy perturbations upon GLUT isoform-mediated emergence of neurodevelopmental and neurodegenerative disorders.

## 1. Introduction

Glucose is the primary energy source for neuronal oxidative metabolism under normal physiological conditions in mammals. A family of facilitative glucose transporters (GLUTs) that transfer glucose down a concentration gradient play a critical role in transporting glucose across lipid bi-layered cellular membranes. In the brain, despite the existence of various other isoforms, GLUT1 and GLUT3 isoforms play a major role in fueling energy requirements [1]. GLUT1 is expressed mainly in the endothelial cells of the blood–brain barrier (BBB), astrocytic processes, and oligodendrocytes found in both gray and white matter [2]. In the endothelial cells of the BBB, GLUT1 is present at both the luminal and abluminal membranes of the endothelial cells, thereby transferring glucose from the blood within the capillaries to the brain interstitium [1]. On the other hand, GLUT3 is located predominantly in pre- and post-synaptic neuronal dendrites and other small neuronal processes, along with axons. In this location, GLUT3 mediates the transport of glucose into high-energy-demanding neurons [2,3]. This is accomplished due to the high affinity of GLUT3 compared to that of the other GLUT isoforms [4]. Owing to its expression in neurons and its abundance in the cortex and cerebellum, GLUT3, more than GLUT1, is considered critically essential for neurotransmission [3,5]. As glucose is the primary energy source for neuronal cells, any condition that lends itself to creating GLUT3 deficiency, as in the case of dietary restrictions, hypoxic–ischemia or significant hypoglycemia, can result in a panoply of neurodevelopmental disorders such as dyslexia, autism spectrum disorder (ASD), epilepsy, and attention-deficit/hyperactivity disorder (ADHD) [6,7,8,9,10]. In addition, there are other conditions that present later in life such as Huntington’s disease (HD) and Alzheimer’s disease (AD), where GLUT3 deficiency is also encountered. In the presence of glucopenia, alternate fuels such as ketone bodies, lactate, and fatty acids are recruited to meet the energy demands of the brain. These substrates are deemed essential in the absence or deficiency of glucose but are not as efficient as glucose in supplying the required cellular energy. Ketones and lactate are transported via the family of monocarboxylate transporters (MCTs), with the MCT1 isoform expressed in the BBB and astrocytes, while MCT2 is found in neurons [11,12]. This is the mechanism behind the amelioration of certain conditions such as epilepsy when a ketogenic diet is introduced therapeutically [13].

GLUT1 and GLUT3 are also expressed as the main isoforms within the early developing embryo and placenta. In the early embryo, in the pre-implantation phase and blastocysts, GLUT1 has been observed primarily in the apical membranes of the trophectoderm and the cells of the inner cell mass [14]. In contrast, GLUT3 is limited in its expression mainly to the basal aspect of the trophectoderm [4]. Absence of either isoform interferes with normal embryonic development resulting in demise, while a deficiency is noted to cause cellular apoptosis [15,16,17]. In addition to these isoforms, GLUT8 has also been observed at this stage and results in a reduction in the litter size rather than complete demise of all embryos, when absent [18].

GLUT1 and GLUT3 are also expressed subsequently in the developing post-implantation embryo and fetus. Embryonic (E) day 7.5 in mice display high expression of GLUT1 and GLUT3 mRNA and protein in extraembryonic tissues rather than the lower amounts in the endoderm [19,20,21]. By E10-10.5, the expression of GLUT1 is confined to the neural tube, gut, heart, and optic vesicle alone, while GLUT3 is limited to the trophectoderm and embryonic gut [19,20,21]. GLUT1 and GLUT3 expression is reduced by E14.5, with GLUT3 mRNA being below the sensitivity of detection [19]. Subsequently, expression of GLUT1 and GLUT3 mRNA and protein is evident in the brain, gradually increasing with advancing gestation to E18 and continuing to increase postnatally (PN), attaining peak concentrations at PN14, and remaining constant thereafter into the adult stage [22]. These observations lend fuel to the importance of GLUT3 in the early stages of embryonic development and subsequent late gestation fetal and postnatal neurodevelopment.

GLUTs’ expression in the placenta mediates glucose transport from mother to fetus via the human syncytiotrophoblastic microvilli or murine labyrinthine region, both of which primarily bear the GLUT1 isoform. In contrast, GLUT3 is noted in cytotrophoblasts of the human and syncytiotrophoblasts of the labyrinthine region, localized on the maternal facing fetal layers of the murine placenta [23,24,25]. In addition, we and others have demonstrated the presence of GLUT8 and GLUT10 in the murine placenta, but their expression is much more restricted [26,27,28]. Given their location and essential function, it is reasonable to connect any changes in the maternal diet or the presence of a metabolic aberration with altered GLUT expression pattern in the placenta, early embryo, and/or the brain of the fetus/offspring. These maternal perturbations set the offspring up for either experiencing fetal growth restriction or ultimately a perturbed neurodevelopmental state underlying altered long-term neurobehavioral outcomes [29,30]. This review focuses first on GLUT3′s expression and function in human and animal models, particularly in the brain as it relates to neurodevelopmental disorders. Second, the focus is on maternal dietary effects and genetic modifications of GLUT3 with a propensity towards developing neurodevelopmental disorders.

## 2. GLUT3

### 2.1. Glucose Transporters (GLUTs and SGLTs Family)

GLUTs are encoded by the *SLC2* gene family and belong to the facilitative superfamily of membrane-spanning glycoproteins [31]. The human GLUT family consists of 14 members, and they are categorized into three classes according to sequence similarity [31,32]. They are further subdivided into three subfamilies, namely class I (glucose transporters GLUT1-4 and GLUT14), class II (fructose transporter GLUT5, GLUT7, GLUT9, and GLUT11), and class III (GLUT6, 8, 10, 12, and H^+^/myo-inositol co-transporter (HMIT)) [33,34,35]. The individual isotypes show different characteristics of substrate specificity, kinetic characteristics, and expression profiles through regulation of their gene expression and they enable tissue/cell-specific monosaccharide (glucose and fructose) uptake [32] (Table 1). GLUT1 was the first membrane transporter detected, purified, and cloned. GLUT1 is expressed in many tissues; as shown in Table 1, the most important biological function in the brain is to transport glucose across the BBB and into cells of glial lineage [31,36,37,38]. GLUT2 is involved in regulation of food intake and the median eminence glucose influx into glucose-sensing cells related to the central nervous system’s regulation of glucose homeostasis [39,40,41,42,43]. GLUT3 is a neuronal glucose transporter due to its exclusive expression in neurons but not glia, including oligodendrocytes or microvascular endothelial cells [2,31]. GLUT4 is expressed in specific anatomic regions of the cerebrum in addition to being present peripherally in insulin-responsive tissues such as muscle and adipose cells, contributing towards maintaining whole-body insulin-regulated glucose homeostasis [44,45,46]. GLUT6 (previously referred to as GLUT9) is also expressed in the brain as well as in white adipose tissue and the spleen [41,42,47,48]. The role of GLUT8 (intracellular) in the brain remains an enigma, but it is perhaps involved in supplying energy to hippocampal neurons [47,48,49,50,51]. GLUT9 is expressed as two differentially spliced variants, GLUT9a and GLUT9b, with each of them being expressed in different tissues and both functioning as a urate transporter [31,52]. GLUT10 is predominantly expressed in the liver and pancreas, although is also observed in the brain, and its gene *SLC2A10* localizes to a chromosomal region previously linked with type 2 diabetes [53,54]. GLUT12 is a glucose transporter widely expressed in the skeletal muscle, adipose tissue, small intestine, and placenta [55].

In addition to the GLUTs, there is another class of glucose transporters, namely the Na^+^/glucose co-transporters (SGLTs). These proteins are a family of membrane-bound glucose transporters that are energy-dependent and function as active transporters. SGLTs transport glucose against a glucose concentration gradient but are dependent on co-transport of sodium. One molecule of glucose is transported with one or two sodium ions which are transported down a sodium concentration gradient [41,56]. These proteins are encoded by the *SLC5* gene with 12 members of a human gene family including 6 SGLTs [56]. SGLT1 is mainly expressed in intestinal mucosa mediating transport of glucose and galactose. In the brain, it contributes to glucose uptake and efflux into certain neurons lining the choroid plexus and periventricular region [56,57]. SGLT2 expression is displayed in the early portion of the proximal renal tubule mediating glucose reabsorption. SGLT2 inhibitors are employed as an adjunct in the treatment of type 2 diabetes mellitus [58,59]. Given its low expression in the cerebellum as well, it has the potential of functioning with SGLT1, although the biological role is questionable at the present time [56]. SGLT3 is not a glucose transporter but rather a glucose sensor in cholinergic neurons and found in skeletal muscle [60]. SGLT4 is primarily expressed in the small intestine and skeletal muscle transporting glucose and mannose. SGLT5 is expressed in the kidney transporting glucose and galactose. Both SGLT4 and SGLT5 are not found in the brain. SGLT6, which is known as the Na^+^/inositol cotransporter (SMIT) 2, is expressed in the brain hypothalamus and substantia nigra. These locations of expression suggest a role for transporting glucose and inositol in the control of food intake and reward processing by recognizing nutrients [57,61,62]. While the members of this family of Na^+^/glucose transporters are biologically important, we will focus mainly on specific members of the GLUT family for the purposes of this review.

**Table 1 nutrients-16-02363-t001:** GLUT and SGLT family.

Classification	Transporters	Km (mM)	Tissue Distribution	Substrate	Function in Brain
GLUT family class I	GLUT1	6.9	Fibroblasts, erythrocytes, brain (endothelial cells, astrocytic processes and oligodendrocytes) >skeletal and cardiac muscle, white adipose tissue, liver	Glucose/Galactose	Transport glucose through BBB and into astrocyte
	GLUT2	11.2	Liver, pancreas, kidney, small intestine, brain (brainstem, thalamus, cortex, hypothalamus, hippocampus) [39,40,43]	Glucose/Galactose /Fructose	Regulation of food and glucose intake
	GLUT3	1.4	Brain (hippocampus, temporal neocortex, gyrus) >testis, placenta, white blood cells, platelets [31]	Glucose/Galactose	Transport glucose to neurons
	GLUT4	4.6	White adipose tissue, skeletal and cardiac muscle cells > brain (basal ganglia, neocortex, hypothalamus, hippocampus)	Glucose	Insulin-dependent regulation of active neuronal circuits
	GLUT14	-	Testis	-	-
GLUT family class II	GLUT5	5	Small intestine >kidney, skeletal muscle, adipose tissue [63,64]	Fructose	-
	GLUT7	0.3	Small intestine, colon [65]	Glucose/Fructose	-
	GLUT9	0.3	GLUT9a: liver, kidney, lung, placenta, leukocytes GLUT9b: kidney, placenta [31,52]	Glucose/Fructose	-
	GLUT11	0.2	Heart, skeletal muscle, pancreas, kidney [66,67]	Glucose/Fructose	-
GLUT family class III	GLUT6		White adipose, spleen, brain (median eminence), peripheral leucocytes	Glucose	Regulation of glucohomeostasis
	GLUT8	2.4	Testis, skeletal muscle, heart, small intestine >brain (amygdala, primary olfactory cortex, dentate gyrus, dorsal hypothalamic area, pituitary stark, posterior pituitary, dentate gyrus, hippocampus)	Glucose	Energy supply for neurons in hippocampus
	GLUT10	0.3	Liver, pancreas > placenta, heart, lung skeletal muscle, kidney, adipose tissue [31]	Glucose/Galactose	-
	GLUT12	4–5	Skeletal muscle, adipose tissue, small intestine [55]	Glucose/Galactose/Fructose	-
	HMIT GLUT13	0.1	Brain (hippocampus, hypothalamus, cerebellum) >white and brown adipose tissues, kidney [68]	Myoinositol	Control of myo-inositol metabolism in brain
SGLT family	SGLT1	0.5	Intestine, Trachea, kidney, heart, testis, prostate, brain (cortical neurons, hippocampal pyramidal cell, Purkinje cells, BBB)	Glucose/Galactose	Removal of glucose from brain interstitium
	SGLT2	5.0	Kidney, brain (cerebellum), liver, thyroid, muscle, heart	Glucose	Not determined
	SGLT3	20	Skeletal muscle, testis, uterus, small intestine, brain (hypothalamus), thyroid	Glucose	Activation of glucosensitive neurons
	SGLT4	2.0	Small intestine, skeletal muscle [56,57]	Glucose/Mannose	-
	SGLT5	-	Kidney [56,57]	Glucose/Galactose	-
	SGLT6	-	Small intestine, brain (hypothalamus and substantia nigra)	Inositol	Recognize nutrients to control food intake and reward processing

### 2.2. GLUT3—Structure and Kinetics

The process of glucose transport as mediated by the family of GLUT proteins, is non-energy dependent and down a concentration gradient from higher to lower glucose concentrations. This process is referred to as facilitative diffusion, with a rate that is more rapid than simple trans-membrane diffusion [69]. All GLUTs demonstrate ≥50% amino acid identity with a similar predicted structure consisting of 12 membrane-spanning α-helical domains with a cytoplasmic amino (N) terminus and a carboxy (C) terminus, with the latter being generally unique to each isoform. The neuronal glucose transporter isoform GLUT3 is encoded by the *SLC2A3* gene and noted within the human chromosome 12p3.3, with a demonstrated higher affinity than other GLUT isoforms [40]. Michaelis Menten’s characteristics depict a low Km of 1.4 mM for GLUT3, which is almost one-fifth of that of the other GLUT isoforms [4,33]. Physiological plasma glucose concentrations are normally maintained between 3.0 and 7.8 mM, whereas glucose concentrations in cerebrospinal fluid (CSF) are much lower around 0.5–2.5 mM [70]. The high affinity of GLUT3 for glucose is critically beneficial in fueling energy-demanding neuronal cells, particularly in circumstances surrounding the availability of lower glucose concentrations than physiologically accepted as normal [71].

### 2.3. GLUT3—Expression/Investigations in Brain and Placenta

GLUT3 mRNAs/proteins are expressed abundantly in mammalian brains as the neuronal glucose transporter [5]. Lower concentrations are expressed in the testis [71,72], spermatozoa, trophectoderm of preimplantation embryo [20,21], placenta [73,74], white blood cells [75], and platelets [71]. Their expression is mostly in the frontal cortex, thalamus, putamen, caudate, hippocampus, and sub-fornical organ located along the antero-dorsal wall of the third ventricle (gray matter-dominant areas), moderately in the cerebellar folia composed of cerebellar white matter, Purkinje neurons and granular neurons (gray matter and white matter mixed areas), and the least in centrum semi-ovale (white matter-dominant area) [10,71,76]. Abundant GLUT3 expression in gray matter, along with much lower levels in white matter, suggests that GLUT3 provides glucose to regions of high metabolic activity [71]. The protein level changes in GLUT3 correspond with synaptogenesis and neuronal maturation in physiologically normal rats and are regulated by the metabolic demand and the regional cerebral glucose utilization rate [1]. In conditions of glucose deficiency, ketones and lactate are used as alternate fuels for energy production. Ketones and lactate traverse the BBB and enter the astrocytes via the monocarboxylate transporter isoform 1 (MCT1) (Km 4.0 mM), while accessing neurons via the MCT2 isoform (Km 0.7 mM) [12] (Figure 1). In addition to the two GLUT isoforms, other isoforms such as GLUT2 [77], GLUT4 [9,78], GLUT6 [42], GLUT8 [78], and GLUT13 [68] are also noted in specific regions or cell types within the brain, albeit in much smaller amounts mediating specific functions.

Homozygous mutations of the GLUT3 gene have only been examined in the context of cell type-specific conditional null mutations in mice [79,80]. This is because of the embryonic demise encountered during the gastrulation phase of development [15]. Such a presentation may hold some validity in humans when early-gestation spontaneous abortion with loss of pregnancy is encountered. Depending on the cell-type selected, namely nestin-, CaMKII- or Emx-1 expressing neural cells, complete absence of GLUT3 has led to varying presentations [79,80]. Employing antisense oligonucleotides specific to the zebra fish GLUT3 gene, which is deployed early in zygotic development, led to significant disruption of brain development with significant growth restriction [81].

Expression of GLUTs in the placenta is highly regulated throughout pregnancy, which includes GLUT3 and GLUT1 isoforms. GLUT1 and GLUT3 are expressed in both the rodent hemotrichorial and human hemomonochorial placentas. In the human, GLUT1 is expressed by the syncytiotrophoblasts that form the microvilli being highly expressed at the internally located basal membrane during the third trimester of pregnancy [82]; likewise, it is mainly expressed in two layers of syncytiotrophoblasts in the labyrinthine region including the junctional region in rodents [25,74]. In contrast, GLUT3 in humans is mainly observed in cytotrophoblasts that are more enriched during the early gestational period, being 48 ± 7% in the second trimester and 34 ± 10% in the third trimester when normalized to 100% of GLUT3 as observed during the first trimester [23]. In the rodent placenta, GLUT3 is expressed in plasma membranes facing the maternal blood supply of the syncytiotrophoblastic layers lining the labyrinthine regions, contributing much more to the total GLUTs concentration [25,74]. This suggests that GLUT3 is important in early gestation of the human pregnancy, due to the presence of actively dividing cytotrophoblasts that impose a higher energy demand than the subsequent fully differentiated syncytiotrophoblasts found more in the second and third trimesters. This early gestation expression of the high-affinity GLUT3 isoform may be necessary for mediating the required glucose transport under an intra-uterine environment of low oxygen and glucose supply prior to the invasion and formation of the intervillous circulation that supplies the early rapidly dividing embryo [24]. In addition to GLUT1 and GLUT3, other isoforms, GLUT4, GLUT8, GLUT9, GLUT10, and GLUT12, are also expressed in the placenta to a limited degree [26,27,28,82].

### 2.4. GLUT3 Is Potentially Associated with Neurodevelopmental and Neurodegenerative Disorders

Given the critical role of neuronal metabolism, aberrant GLUT3 expression due to pathological conditions of gene copy number variants (CNVs) or gene mutations, creates the conditions for the emergence of neurodevelopmental disorders. During normal murine development, both GLUT1 and GLUT3 proteins are expressed at low levels in the embryonic and early postnatal brain, only to reach peak concentrations at PN day 10 to 15 [22]. During this phase, hypothyroidism is known to alter expression of these isoforms with a detrimental effect on the postnatal brain [83]. Sp1/Sp3 and phosphorylated CREB were demonstrated to regulate transcription of murine brain GLUT3 gene [84,85]. GLUT3 protein expression increases are paralleled by the corresponding mRNA levels but not associated with the increases in transcriptional rates, whereas GLUT1 protein increase coincides with mRNA levels [22]; however, the processing has been shown to be post-transcriptional [86]. These data suggest that GLUT3 gene is regulated either post-transcriptionally/translationally [22] or there is a lag between transcription and post-transcriptional/translational events. Thus, employing the transgenic GLUT3-luciferase mouse model, we deciphered the developmental timing of transcription to precede the peak levels of brain GLUT3 mRNA and protein [87]. In addition, prior studies have revealed transcriptional control of GLUT3 by various trans-acting nuclear factors [84,85], supporting the idea that transcription occurs before the timing of post-transcription/translation in the case of this gene. Also, other groups have reported that untranslated gene regions carrying mutations have the potential to affect neurodevelopment [88]. These are important to unravel before investigating the impact of either genetic modifications or dietary perturbations in the mother during pregnancy or lactation.

Hypoxic–ischemia (which involves pan-nutritional deficiency including an oxygen-deficient state) early in development or subsequently during juvenile or adult stages enhances brain GLUT3 expression during the early energy-demanding recovery period but is rapidly reduced in the cortex and thalamus, reflecting apoptotic and necrotic injury to the brain during this early response (24 h). The later response (72 h) results in extensive regional area necrosis [89]. Heterozygous GLUT3 null mice with hypoxic–ischemic brain injury is reported to show more severe cellular apoptosis and necrosis than wild-type mice and tend to present with spontaneous clinical seizures [76]. Further, exposure to pentylenetetrazole caused a longer duration of seizures in these genetically modified adult mice [76]. These results suggest that GLUT3 could underlie an adaptive response to hypoxic–ischemia, providing some insights into the development of a therapeutic intervention targeting activation of this isoform in neurons.

In a prior study, pentylenetetrazole- and kainic acid-induced seizures increased both GLUT1 and GLUT3 expression in rat brains [90]. Another group reported that lithium–pilocarpine-induced seizures also increased brain GLUT3 expression, albeit only transiently [91]. Leroy et al. reported on the upregulation of MCT1 and MCT2, suggesting the possibility of ketone utilization as an alternate fuel in the presence of epilepsy [91].

Disruption of glucose metabolism in neuronal cells underlies the pathogenesis and progression of Huntington’s disease (HD) [92,93,94]. In humans, GLUT3 expression was significantly decreased in an advanced stage of HD, while there was no significant difference compared to normal controls in the early stage of the disease despite brain glucose utilization being reduced [95,96]. Li et al. elucidated that Rab11, a Rab family GTP-ase involved in cellular endosomal recycling, influences brain glucose metabolism in 140 CAG repeat knock-in mice [97]. Additionally, they demonstrated that in the presence of increased gene *SLC2A3* CNVs, which led to increasing concentrations of the GLUT3 protein in HD patient’s neural cells, the age of onset in acquiring HD clinical features could be delayed [98]. In another neurodegenerative disorder, Alzheimer’s disease (AD), reduced GLUT1 and GLUT3 expression has also been reported in the human brain [99]. Liu et al. observed a concordance between brain GLUTs and other established AD markers; GLUT3 decreases in correlation with the decrease in O-GlcNAcylation and the appearance of hyperphosphorylation of the tau protein in the brain [100]. An et al. demonstrated lower brain GLUT3 expression with an associated greater severity of the amyloid plaque and neurofibrillary tangles development, along with increasing severity of the clinical symptoms in AD patients [101]. Importantly, recent studies have emphasized a possible connection between neurodevelopmental and neurodegenerative disorders [102]. For example, in HD, aberrant neurodevelopment sets the stage for future neuronal dysfunction and degeneration [103,104]. Similar findings have been reported for AD [105]. It is thus possible that in both types of disorders, GLUT3 deficiency during brain development plays an important role. Several clinical studies reported that patients presenting with symptoms of attention deficit hyperactivity disorder (ADHD) reveal CNVs of a locus in the *SLC2A3* gene, with the presence of *SLC2A3* gene duplication being associated with altered working memory and cognitive response control, which are features of ADHD [6,106]. Reduced mRNA expression of *SLC2A3* in white blood cells was reported to be negatively associated with a late left-lateralized auditory mismatch [8]. Skeide et al. reported that these SLC2A3 gene variants were associated with a phonological awareness phenotype connected to dyslexia [107]. Heterozygous GLUT3-deficient (GLUT3^+/−^) mice presented with autism spectrum disorder (ASD)-like clinical features and neurobehaviors, namely increased electroencephalographic seizure activity, abnormal spatial learning and working memory, reduced vocalizations, and stereotypies [10]. On the other hand, homozygous GLUT3-deficient mice resulted in embryonic loss at E6.5 due to the critical role of GLUT3 in the trophoectoderm of the early embryo [15,108,109]. However, when conditional null homozygous mutations were created to overcome embryonic demise using the nestin-Cre transgene driving the floxed Slc2A3 gene mutation, no changes were evident until after PN day 10, after which significant growth restriction with microcephaly emerged with demise either before or after weaning from the mother [80]. On the other hand, employing the Emx1-Cre or CaMKII-Cre led to survival into adulthood with successful reproduction. The former demonstrated some features akin to ADHD, whereas the latter expressed lack of fear for spatial exploration or social novelty with some disruption of spatial cognition [79]. These mouse models are essential in teasing out the role of GLUT3 in specific neuron subtypes, thereby affecting different neurobehavioral functions.

## 3. Maternal and Postnatal Dietary Exposures Affect Offspring’s GLUTs

### 3.1. High-Fat Diet

Obesity, with a body mass index (BMI) of 30 or higher, is a global issue, although it is more predominant in Western countries. In the United States, about one-third of reproductive-age women are diagnosed with obesity due to their dietary habits and sedentary lifestyle [110,111]. Many studies have shown that maternal obesity causes long-lasting adverse events in the offspring, such as neurodevelopmental disorders, metabolic derangements, and associated cardiovascular diseases [112,113,114,115]. In addition, maternal complications ensue such as gestational hypertension, pre-eclampsia, and gestational diabetes mellitus [116,117]. Recent investigations employing nonhuman primate and rodent models of maternal obesity due to consumption of a high-fat diet demonstrated an increase in inflammatory cytokines, insulin, fatty acids, and triglycerides in the maternal circulation that affect the fetal homeostatic milieu, thereby perturbing the fetal brain melanocortinergic, serotoninergic, and dopaminergic neural networks [30,112,118]. These aberrations acquired in utero are not ameliorated upon postnatal introduction of a regular chow diet, supporting the concept of fetal mal-programming, due to maternal obesity-associated systemic alterations [119]. Upon examination of placental GLUTs, various groups of investigators, including our group, previously demonstrated that wild-type murine placental GLUT1 increases during late gestation when exposed to pre-conceptional and gestational high-fat diet, with no concomitant change in GLUT3 observed [120,121]. While Ganguly et al. observed no change in the immediate postnatal suckling litter body weights [120], Jones et al. reported an increase in body weight gain in response to hyper-caloric high-fat exposure [121]. The difference between the two studies rested on the energy emanating from the dietary carbohydrate composition, being double in the latter compared to the former (52% vs. 27%) [120,121]. Rosario et al. demonstrated an increase in murine placental GLUT1 and GLUT3 concentrations in response to consumption of a high-fat and high-sucrose diet pre-conceptionally and gestationally, resulting in increased postnatal body weights [122]. In contrast to these studies in wild-type mice, a heterozygote null mutation of the Slc2A3 gene with 50% of GLUT3 protein, failed to reduce intra-placental and transplacental glucose transport [123]. Pre-gestation and gestational consumption of a high-fat diet by GLUT3^+/−^ mice increased placental GLUT1 and GLUT3 concentrations (emanating from a single allele), with subsequent postnatal litter body weight increase [120]. Collectively, these studies suggest that while GLUT1 may mediate intra-placental glucose transport, GLUT3 mediates trans-placental glucose transport from mother to fetus, impacting the ultimate weight gain of the offspring.

Supplemental *n*-3 fatty acid in preterm neonatal pigs ameliorates brain immunohistochemical changes by reducing apoptotic cells in the internal granule cell layers [124]. Despite the multiple beneficial effects regarding the intake of *n*-3 fatty acid-rich diets [125,126,127], one must be aware of the dose-dependent deleterious effects as well, particularly if this exposure is during pregnancy or lactation. We had previously reported that rats on an *n*-3 fatty acid-enriched high-fat diet during lactation, subsequent to maternal *n*-6-enriched high-fat food intake during gestation led to microcephaly with decreased brain GLUT3 concentrations with no difference in GLUT1 [128]. Stachowiak et al. also reported that embryonic day 21 mice exposed to a maternal high-fat diet consisting most probably of *n*-6 fatty acids demonstrated a diminution in migration and maturation of periventricular stem cells within the fetal brain [129]. Thus, despite the reported benefits seen in the metabolic milieu with *n-3* fatty acids, these studies indicate caution when it comes to neurodevelopment.

### 3.2. Ketogenic Diet

A ketogenic diet has been established as a therapeutic alternative for anti-epileptic drug (AED)-resistant epilepsy [13,130,131,132]. Most often, there is either a germline or somatic genetic mutation basis for such epilepsies, warranting surgical resection in the case of focal cortical dysplasias (FCDs). More often, as in the case of the germline Slc2A1 gene mutation (haplodeficiency), infantile-onset epilepsy with persistently low cerebrospinal fluid glucose concentrations, delayed development, microcephaly with intellectual disability, and complex movement difficulties which include gait abnormalities are encountered and referred to as GLUT1 deficiency syndrome [133,134]. In this specific case, while AEDs are ineffective, a ketogenic diet encompassing a high-fat and low-carbohydrate diet, has been effective in controlling seizures. The ketogenic dietary intervention induces a metabolic state of ketosis by increasing the exogenous supply of ketone bodies (β-hydroxybutyrate, acetoacetate, and acetone) and enhancing the endogenous production by fatty acid oxidation. Overall, a 50% or greater reduction in seizure frequency after three or six months of a ketogenic dietary intervention in children has been reported [135,136,137]. The mechanism of amelioration rests on the provision of a non-glucose alternate source of energy for fueling the metabolism in neurons, restoring impaired mitochondrial function, regulating neurotransmission at pre- and post-synaptic junctions (changing the amount of GABA, glutamate, and adenosine receptors), having anti-inflammatory effects on neurons and reducing the amount of reactive oxygen species, along with having epigenetic effects on DNA and histones [13,131]. The side effects of a ketogenic dietary therapy consist of dehydration, weight loss, and hypertriglyceridemia arising as common early-onset complications, along with osteopenia, renal stones, and cardiomyopathy, setting in as late-onset complications [133,138,139,140]. Besides epilepsy, the ketogenic diet has been introduced for treatment of other neurological disorders such as Parkinson’s disease, multiple sclerosis, AD, and migraine [131]. Autistic behaviors have also been reported to potentially experience amelioration in response to a ketogenic diet [141].

In contrast, isolated GLUT3 deficiency due to a gene mutation has only been reported so far in two children presenting with myelomeningoceles [142]. Thus, not much is known in humans regarding conditions associated with haplodeficiency of GLUT3. Most of the studies thus far have occurred in animal models with genetic modifications. Cheng et al. reported a drastic intervention of a 90% calorie restriction with intake of a ketogenic diet over a period of 7 days which increased rat brain GLUT1 and GLUT3 mRNA expression when compared to the control diet with 90% calorie restriction alone [143]. Classical GLUT3^+/−^ mice presented with a neurodevelopmental disorder akin to ASD as suggested by neurobehaviors, such as increased electroencephalographic seizure activity, abnormal spatial learning and working memory, cognitive inflexibility, stereotypic behaviors of low frequency, abnormal socialization, and reduced vocalizations. Otherwise, homozygous GLUT3 knock-out mice could have pregnancy loss [10,15]. GLUT3^+/−^ mice revealed an earlier developmental increase in neuronal MCT2 expression at PN 14 through PN60 than that encountered in wild-type mice [10]. Gestational and postnatal ketogenic dietary intervention in the case of these GLUT3^+/−^ mice improved electroencephalographic seizures and abnormal sociability but failed to affect abnormalities in spatial learning, working memory, cognitive inflexibility, and reduced vocalization, suggesting only a partial amelioration of the neurobehavioral presentation by enhanced utilization of ketones as an energy fuel when neurons experience glucopenia [144,145]. Such partial amelioration attests to the successful transfer to fetus of maternal ketones that build up when exposed to a ketogenic diet [145]. Unlike these reports, exposure to a ketogenic diet in normal wild-type mice can prove detrimental to murine neurodevelopment [144]. In addition, Sussman et al. described maternal infertility subsequently when ketoacidosis ensued during the lactation phase. What is most concerning is the ultimate change in the postnatal brain structure encountered in response to prenatal and early postnatal (before weaning) exposure to the ketogenic diet in the normal state. Such an exposure decreased the volume of the cortex, hippocampus, corpus callosum, and lateral ventricle, with relative enlargement of the hypothalamus and medulla at PN11.5 and PN21.5 [146]. These changes persisted into later adult life, where differences in brain morphology were associated with behavioral alterations such as affecting susceptibility to anxiety and depression with elevated hyperactivity, despite reverting to a normal diet post-weaning [147]. Such reports raise the question of whether a ketogenic diet during pregnancy and lactation can program the offspring’s neurodevelopment. While extrapolation to the human may be premature, it raises concerns and suggests steering away from a ketogenic diet during pregnancy.

On the other hand, activating *CNVs* in the GLUT3 gene have been reported in individuals with ADHD [6,106], and while no activating GLUT3 mutation murine models exist or have been studied so far, it is imperative to determine this clinical association in a cause-and-effect manner.

### 3.3. Energy Restriction

Inadequate maternal calorie intake causes intrauterine growth restriction (IUGR), which describes a condition of inadequate fetal growth rate that prevents infants from reaching their optimal growth potential [29,148,149]. Etiologies of IUGR may be due to multiple causes with various detrimental outcomes; however, aberrant neurodevelopment is a major risk factor. IUGR infants are known to develop adverse neurodevelopmental outcomes such as neurodevelopmental delay, behavioral problems, motor and cognitive deficiencies, and cerebral palsy [148,150,151,152,153,154,155,156,157,158]. Genetic mechanisms for neurodevelopmental disabilities with IUGR are suggested [159,160,161,162], but remain to be elucidated. Thus, an explosion of animal models to examine the state of IUGR has occurred. Iskusnykh et al. reported that IUGR pigs showed reduced cerebellar weight and the number of granule cells and the JamC/Pard3a molecular pathway mediates compromised initiation of granule cellular radial migration, which contributes to the cerebellar pathology [163].

In the case of placental GLUTs, there is some variability in the results depending on the model and severity of the IUGR state. However, studies have also emanated in the post-parturient human placenta, where increased placental GLUT3 with no change in GLUT1 was observed in the late gestation presentation of a milder form of IUGR [164]. In contrast, Kainulaineu et al. observed no change in human placental GLUTs in a moderately severe form of IUGR [165]. In a large animal model of epitheliochorial placentas such as ewes, when 50% nutrient restriction was imposed during the first half of gestation, increased placental GLUT1 and GLUT3 mRNA expression was noted during mid-gestation [166]. Das et al. reported that late gestational chronic hypoglycemia induced by insulin infusion in sheep, rather than calorie restriction, did not change placental GLUT3 expression but decreased GLUT1 with no difference in transplacental glucose uptake [167]. In small animal models with hemochorial placenta, such as mice with 80% nutrient restriction from early gestation onwards throughout pregnancy, increased placental GLUT1 expression with no change in fetal glucose clearance [168]. In contrast, mid-to-late gestation dietary restriction in mice decreased placental GLUT3, with no change in GLUT1 expression and maternal–fetal glucose transport [169,170]. Comparing mild (~25%) and moderate (~50%) calorie restriction only during mid-to-late gestation led to a differential response, where the mild condition increased and moderate condition decreased placental GLUT3 with a concomitant effect on transplacental glucose transport [123]. Though maternal dexamethasone administration is commonly used to improve mortality and critical morbidity in preterm neonates, it is known to reduce birth weight and multiple dosing regimens of administration in animal models (unlike dosing in human pregnancies) are used to induce IUGR [171,172,173]. Late-gestation repetitiously dosed dexamethasone-induced IUGR in mice also decreased placental GLUT1 and GLUT3 [174]. Collectively, what is apparent is that the placental GLUTs are reliant on maternal diet, and its effect is dependent on the gestational timing and severity of IUGR. All these factors play a role in the adaptive change in placental GLUTs, which is required for maintaining transplacental glucose transport, towards fueling fetal survival at the expense of fetal growth.

Brain GLUTs are also affected by IUGR. A rat IUGR model consisting of maternal uterine artery ligation causing uteroplacental insufficiency increased fetal brain GLUT1 protein concentrations which persisted until PN21 [175].

Dietary based energy-restricted mice during mid-to-late gestation (~50%) reduced fetal brain GLUT3 protein concentrations by 50% along with a reduction in brain serotonin (5-HT) and serotonin transporter (SERT) concentrations [176]. In addition to changes in brain GLUTs and an imbalanced 5-HT-SERT axis, a reduction in cortical thickness and microcephaly was encountered [177]. In addition, a reduction in nestin (progenitor marker), β-III Tubulin (immature neurons), GFAP (astrocyte marker), and O4 (oligodendrocyte marker) -expressing cells emerged. Maternal dietary restriction has the potential for disturbing neural progenitors’ differentiation into cerebral cortical brain cells [177]. Further, the postnatal offspring of the dietary-restricted rats displayed an adaptive increase in brain GLUT1, GLUT3, and brain-derived neurotrophic factor with a concomitant reduction in GLUT4 protein concentrations [9,178] (Table 2). Of importance is the emergence of perturbed neurobehaviors subsequently during adult life such as heightened anxiety and some cognitive impairment [9]. Further, during aging, these rat offspring demonstrated reduced brain IRS-2 and pAkt and increased GFAP and amyloid-p42, akin to that observed in AD. Imposition of postnatal calorie restriction on IUGR, while considered therapeutic in controlling metabolic derangements, is highly detrimental to brain development. While most of the studies described here involve calorie restriction, selective protein restriction and other micronutrient deficiencies have also been shown to affect neurodevelopment adversely [9,178].

In addition to dietary restriction, energy deprivation also occurs under hypoxic–ischemic conditions. We have previously observed that 45 min of hypoxic exposure followed by carotid arterial ligation-induced ischemia led to GLUT3^+/−^ more than wild-type mice incurring brain nuclear pyknosis and cellular necrosis. These GLUT3^+/−^ mice were also observed to display clinical seizures or non-symptomatic EEG seizures alone, while the wild-type mice had no seizures [76]. In vitro studies demonstrated a role for HIF-1α and CREB in inducing GLUT3 expression in response to hypoxia in murine neuroblastoma cells, and in vivo studies in mice corroborated these findings [179]. These reports assign a protective role for neuronal GLUT3 against hypoxic–ischemia, since a reduction in neuronal GLUT3, which worsens brain injury in response to hypoxic–ischemia, is seen as increased cellular apoptosis and necrosis. To determine the relevance to human infants, we undertook a study employing blood cells in hypoxic–ischemic newborn infants subjected to therapeutic hypothermia. Hypoxic–ischemia led to changes in red cell GLUT1 but not in white blood cell GLUT3 [180]. What is important to note is that at this age, while red blood cell GLUT1 concentrations are high, white blood cell GLUT3 concentrations are low [75]. Thus, such studies later in development when white blood cell GLUT3 concentrations are much higher are likely to yield differing results in regard to the role played by both GLUT1 and GLUT3.

## 4. Conclusions

GLUT3 is a high-affinity glucose transporter that plays an important role in the brain and placenta, transporting the primary fuel, namely glucose, to neuronal cells or trans-placentally to the fetus, respectively. Malfunction or deficiency of GLUT3 has the propensity to interfere with sustenance of a pregnancy and ultimately result in intrauterine growth restriction and neurodevelopmental disorders. It also has the capacity to alter the clinical course of adult-onset disorders such as HD and AD. The ketogenic diet has the potential of serving as an alternate fuel sustaining cellular energy metabolism in the presence of a dysfunctional GLUT1 or GLUT3, whether in response to dietary changes or genetic mutations. However, it is imperative to develop newer therapeutic modalities to address these conditions that interfere with the well-being of the mother and developing fetus during pregnancy and the postnatal development of these disorders, with a higher risk of developing aberrant neurobehavioral and subsequent age-related disorders.

## Figures and Tables

**Figure 1 nutrients-16-02363-f001:**
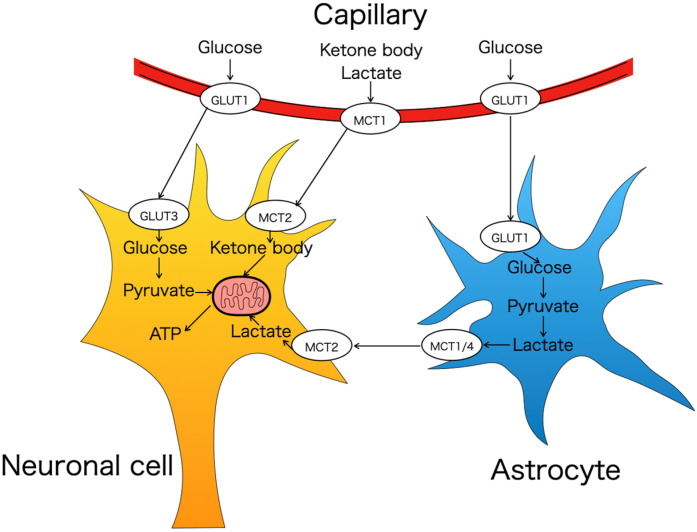
The transport mechanisms underlying transfer of glucose, ketone body, and lactate across the blood–brain barrier from capillary vessels to neurons and astrocytes. GLUT; glucose transporter, MCT; monocarboxylate transporters.

**Table 2 nutrients-16-02363-t002:** Effect on GLUTs expression in placenta and brain and neurodevelopment by maternal dietary change.

		Brain GLUTs Expression and Neurodevelopment	Placenta GLUTs Expression
High-fat diet	Maternal	- Reduced GLUT3 expression (rat) [129]. - *n*-3 fatty acids improve vision development (human) [126,127]. - No significant change in the first 2 years of age, but significant positive effects in cognitive tests 3–6 years of age (human) [125,126,127].	- Increased GLUT1, no change GLUT3(mouse) [120,121]. - High-fat + high-sucrose diet increased GLUT1 and GLUT3 (mouse) [122]. - GLUT3^+/−^ mice increased GLUT1 and GLUT3 [120].
	Postnatal	- Exposure to postnatal *n*-3 fatty acid-enriched high-fat diet after maternal *n*-6 fatty acid-enriched high-fat diet reduced GLUT3, with no change in GLUT1 (mouse) [128].	-
Ketogenic diet	Maternal	- Brain structural change (decreased the volume of cortex, hippocampus, corpus callosum) (mouse) [146]. - Reduced susceptibility to anxiety and depression, elevated hyperactivity (mouse) [147].	- No data regarding GLUTs.
	Maternal and Postnatal	- GLUT3^+/−^ mice decreased ASD-like behaviors and decreased sociability in wild-type mice [144].	-
Energy restriction	Maternal	- Maternal uterine artery ligation in late gestation decreased GLUT1 (rat) [175]. - 50% dietary restriction during mid-to-late gestation decreased fetal brain GLUT3 with reduction in brain 5-HT and SERT concentrations with anxiety and reduced cortical thickness (mouse) [176,177].	Variable results depending on the model and severity of the IUGR state [164,165,166,167,168,169,170,174]. - Late-gestation mild-form IUGR showed increased GLUT3, with no change in GLUT1 (human) [164]. - Moderately severe-form IUGR showed no change in GLUT3 and GLUT4 (human) [165]. - 50% nutrient restriction during the first half of gestation increased GLUT1 and GLUT3 during mid-gestation (ewe) [166]. - Chronic hypoglycemia by insulin decreased GLUT1, with no GLUT3 change (sheep) [167]. - 80% nutrient restriction throughout pregnancy increased GLUT1 (mouse) [168]. - Mid-to-late gestation 25% dietary restriction increased GLUT3, and 50% restriction decreased GLUT3 (mouse) [169,170]. - Late gestation dexamethasone-induced IUGR decreased GLUT1 and GLUT3 (mouse) [174].
	Postnatal	- Offspring of 50% dietary-restricted rats displayed an adaptive increase in GLUT1, GLUT3, and BDNF with heightened anxiety and cognitive impairment (rat) [9,178].	-

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
