# Peer review of "Neurodevelopment Is Dependent on Maternal Diet: Placenta and Brain Glucose Transporters GLUT1 and GLUT3"

_nutrients, 2024, doi:10.3390/nu16142363_

Round 1

Reviewer 1 Report

Comments and Suggestions for Authors

This is an interesting paper. Nevertheless,several aspacts should be considered.

Major:

L. 2f: The title should be changed as the paper doen't only address GLUT3 in brain or the paper must be re-focused.

L. 81ff: This para should be shortened, and the body of the text transformed into a table, with symbols of low or high expression, the body of the text more for general explanations and biological concepts. It would be very helpful to address aspects of fetal developmental changes as well.

L. 362: This sentence is misleading, as energy restriction frequently is essential not to gain weight over normal values during pregnancy. Could it be that the authors mean energy restriction to values below the needs for fetal growth and maternal needs?

L. 385: Comment the clinical impact here, as systemic glucocorticoids are clinical routine in the management of preterm delivery.

L. 394ff: These animal models are critical and frequently do not match the human/clinical situation. Moreover, 50% of what? Energy, Protein? What is the clinical impact here? It isn't an adequate model for the clinical situation of small for gestational age infants.

Table 1: The data of the table should be part of the main body rather than only of the conclusion.

Minor:

L95: Remove "of the liver" or write: liver parenchymal cells.

L116: What means "susceptibility to be linked to"? Please explain.

L 119-122: Grammar of sentence; meaning unclear.

L. 134: Delete, as described belor or move Km values here to save text volume.

L. 139f: Values should be integrated into a table, like that suggested above.

L. 141: Upper values are wrong. Postprandial values may exceed 5.6mM under non-pathological conditions.

L. 142: When plasma values are 3.0-5.6mM, then 0.5-2.5 isn't a narrow range. By contrast, isn't it the larger range for which a low Km is required to achieve glucose transport at low concentrations?

L. 159-162: Please specify this with respect to Km values, distribution in tissues and transfer rate as for GLUT3.

L. 169-171: What do the percentages mean? Of 1st trimeter expression or relative to all GLUT expression? Is there an absolute decrease or a relative decrease?

L. 178: What does low oxygen have to do with low glucose, except if in early gestation energy supply is partly achieved by glycolysis?

L 180f: Specify other GLUTs, please. Provide mechanistic explanations for them, if possible. AS it is, the information is worthless.

L. 188: “10 to 15”: which species is meant?

L. 192: “whereas”: Don’t understand the logic of sentence.

L. 260f: Is this in rodents or proven in humans as well? Specify!

L. 270: “high fat exposure”: Replace by hypercaloric high fat diet/exposure. As it is, it’s partly misleading as it possibly isn't the fat alone, but the hypercaloric diet and the fat vs. regular chow diets. High fat diets induce adiposity in rodent experiments, but high fat fraction in humans doesn't necessarily do that.

L. 403: “offsprings2 rather than “offspring”.

L 418f: grammar of sentence.

Author Response

Responses to Reviewers’ Comments

Reviewer 1:

Comment: This is an interesting paper. Nevertheless, several aspects should be considered.

Response: Thank you. Please see below regarding how we have addressed the comments.

Major:

Comment: L. 2f: The title should be changed as the paper doen't only address GLUT3 in brain or the paper must be re-focused.

Response: →L 3: We have changed the title to “Glucose Transporters GLUT1 and GLUT3” as suggested by this reviewer. Although we also mention, very briefly, other transporters, the main focus of the review is on GLUT1 and GLUT3.

Comment: L. 81ff: This para should be shortened, and the body of the text transformed into a table, with symbols of low or high expression, the body of the text more for general explanations and biological concepts. It would be very helpful to address aspects of fetal developmental changes as well.

Response: →L 88: We created Table 1 as a substitute to this paragraph and changed the description to contain only GLUTs expressed in brain and placenta. In addition, in response to Reviewer 2’s comment on this same paragraph, we have now added SLGTs to the paragraph and included the GLUTs distribution and function in brain within this same table.

Comment: L. 362: This sentence is misleading, as energy restriction frequently is essential not to gain weight over normal values during pregnancy. Could it be that the authors mean energy restriction to values below the needs for fetal growth and maternal needs?

Response: →L 388: We added the word “inadequate” in the sentence. We often see mothers with self-body image problems, the worst being anorexia/bulimia, or women on drugs or being depressed, who do not display adequate caloric intake. Also on the global platform, under resourced countries have women consuming less calories than required especially when mired in poverty.

Comment: L. 385: Comment the clinical impact here, as systemic glucocorticoids are clinical routine in the management of preterm delivery.

Response: →L 416: We added a sentence regarding maternal dexamethasone administration.
Most clinical studies have been focused on lung development or head circumference with developmental outcomes. Instead, studies in animal models have used steroids repeatedly to create IUGR in the fetus. Thus, it is difficult to compare them as a preventative measure, but rather in creating IUGR.

Comment: L. 394ff: These animal models are critical and frequently do not match the human/clinical situation. Moreover, 50% of what? Energy, Protein? What is the clinical impact here? It isn't an adequate model for the clinical situation of small for gestational age infants.

Response: →L 427: We added the word “ supplied energy” in the sentence. While we appreciate this reviewer’s input, we are aware that on the global platform, the cause for IUGR is poverty and inadequate consumption of calories by women during pregnancy. In well resourced countries there are other reasons such as hyperemesis gravidarum, depression, body image issues, etc. that do lead to reduced intake of required calories causing IUGR. Besides our group has extensively characterized this calorie restricted model in mice and rats and noted particularly in the former that changes in uterine blood flow mimic uteroplacental insufficiency which gradually occur over the period of gestation mimicking the human condition of IUGR closely (Reference: Ganguly et al. Maternal calorie restriction causing uteroplacental insufficiency differentially affects Mammalian placental glucose and leucine transport molecular mechanisms. Endocrinology 2016, Oct; 157(10):4041-4054. Doi:10.1210/en.2016-1259. PMCID:PMC5045505).

Comment: Table 1: The data of the table should be part of the main body rather than only of the conclusion.

Response: →L 444: We moved this table into the main body. We also changed the name to “table 2” as we have introduced a new table 1.

Minor: 

Comment: L95: Remove "of the liver" or write: liver parenchymal cells.

Response: → L 74: This was regarding GLUT2. We removed this description to shorten this paragraph, as it is not directly related to the brain.

Comment: L116: What means "susceptibility to be linked to"? Please explain.

Response: →L 111: We have now mentioned the location within the chromosomal region in this sentence.

Comment: L 119-122: Grammar of sentence; meaning unclear.

Response: → L 113: This was regarding HMIT, we removed this sentence to shorten this paragraph and place this information in table 1 (The sentence removed is: H+/myo- inositol co-transporter, HMIT (GLUT13), SLC2A13 transcripts are expressed predominantly in the brain, with significantly higher expression in the hippocampus, hypothalamus, cerebellum, and brainstem and at a relatively low level in white and brown adipose tissues and kidney [52].).

Response: L. 134: Delete, as described below or move Km values here to save text volume.

Comment: → L 145: We deleted the sentence “ The structural characteristics of GLUT3 responsible for inducing a high affinity for glucose are the level of N-glycosylation of the first extracellular loop, the cytoplasmic loop between trans-membrane (TM)7 and TM8, and the amino acid residues near the constructed pore between TM9 and plasma membrane (PM)11 domains, which bind hydrogen in the glucose moiety [4,7].” Also, the Km values were moved into table 1.

Comment: L. 139f: Values should be integrated into a table, like that suggested above.

Response: →L 146: We moved Km values of GLUT2-4 into table 1.

Comment: L. 141: Upper values are wrong. Postprandial values may exceed 5.6mM under non-pathological conditions.

Response: → L 148: In response to this comment we changed the value “7.8 mM” (= 140 mg/dl postprandial value).

Comment: L. 142: When plasma values are 3.0-5.6mM, then 0.5-2.5 isn't a narrow range. By contrast, isn't it the larger range for which a low Km is required to achieve glucose transport at low concentrations?

Response: We deleted the phrase “within a narrow range”.

Comment: L. 159-162: Please specify this with respect to Km values, distribution in tissues and transfer rate as for GLUT3.

Response: → L 167: We have now added MCTs’ Km values.

Comment: L. 169-171: What do the percentages mean? Of 1st trimester expression or relative to all GLUT expression? Is there an absolute decrease or a relative decrease?

Response: →L 193: Yes, there is a relative decrease compared to GLUT3 in 1st trimester, when normalized and expressed as 100%. We have changed the description.

Comment: L. 178: What does low oxygen have to do with low glucose, except if in early gestation energy supply is partly achieved by glycolysis?

Response: →L 201: GLUT3 does not require oxygen to transport glucose into the embryo. Understood that there is low oxygen during implantation of the embryo with a heightened need for glucose to supply the required energy (there is no provision of oxygen and glucose). However, under conditions of hypoxic-ischemia, there is general nutrient restriction due to the ischemia, and oxygen is also considered an essential nutrient, so hypoxia is considered restriction of a nutrient by some in the nutritional field.

Comment: L 180f: Specify other GLUTs, please. Provide mechanistic explanations for them, if possible. AS it is, the information is worthless.

Response: → L 169: GLUTs (GLUT4, 8, 9, 10, and 12) are specified, but we think the mechanistic explanation for each and every GLUT and then the SGLTs is far beyond the scope and will add too much length to this review article.

Comment: L. 188: “10 to 15”: which species is meant?

Response: → L 212: It is mice. We specified it in the sentence.

Comment: L. 192: “whereas”: Don’t understand the logic of sentence.

Response: → L215: We have changed the whole sentence relating to this phrase.

Comment: L. 260f: Is this in rodents or proven in humans as well? Specify!

Response: → L 291: It is only animal models. We mentioned “non-human primate and rodent” in the sentence.

Comment: L. 270: “high fat exposure”: Replace by hypercaloric high fat diet/exposure. As it is, it’s partly misleading as it possibly isn't the fat alone, but the hypercaloric diet and the fat vs. regular chow diets. High fat diets induce adiposity in rodent experiments, but high fat fraction in humans doesn't necessarily do that.

Response: → L 302: We added the phrase “hyper caloric”.

Comment: L. 403: “offsprings2 rather than “offspring”.

Response: → L 437: While “offsprings” are in general use, it is wrong as a plural noun, “offspring” denotes both the plural and singular noun (Cambridge Dictionary).

Comment: L 418f: grammar of sentence.

Response: → L 454: We added “which” to correct the grammar of this sentence.

In summary, we appreciate suggestions of both reviewers and hope it has significantly strengthened the presentation of our review.

Reviewer 2 Report

Comments and Suggestions for Authors

In this manuscript, the authors attempted to review the role of diet/energy perturbations upon GLUT isoform-mediated emergence of neurodevelopmental and neurodegenerative disorders. Here are my suggestions to improve the manuscript:

1) In this review authors need to add the description of the role of sodium-dependent glucose transporters (SGLTs). The table describing different isoforms of GLUTs, SGLTs, their functions, and distribution in the brain will strengthen the paper.

2) The authors are encouraged to discuss other genetic and molecular mechanisms causing various forms of neurodevelopmental disabilities related to calorie/energy restriction during prenatal and postnatal stages. Also, I believe that a discussion of nutrition components improving neurodevelopment will strengthen the paper.  As a reference: PMID: 32718081, PMID: 37961498,  PMID: 33259808.

Author Response

Responses to Reviewers’ Comments

Reviewer 2:

Comment: In this manuscript, the authors attempted to review the role of diet/energy perturbations upon GLUT isoform-mediated emergence of neurodevelopmental and neurodegenerative disorders. Here are my suggestions to improve the manuscript:
Response: Thank you for your review. Please see below regarding how we have addressed the comments.

  • Comment: In this review authors need to add the description of the role of sodium-dependent glucose transporters (SGLTs). The table describing different isoforms of GLUTs, SGLTs, their functions, and distribution in the brain will strengthen the paper.

Response: → L 114: We added SGLTs as recommended by this reviewer. Also, we have now created a new table 1 (L 135) which includes SGLTs with their function and distribution in brain.

  • Comment: The authors are encouraged to discuss other genetic and molecular mechanisms causing various forms of neurodevelopmental disabilities related to calorie/energy restriction during prenatal and postnatal stages. Also, I believe that a discussion of nutrition components improving neurodevelopment will strengthen the paper.  As a reference: PMID: 32718081, PMID: 37961498, PMID: 33259808.

Response: → We appreciate your suggestions. We added the other molecular and genetic mechanisms L393-L398, with reference to the recommended paper Iskusnykh et al., 2021. Regarding genetic mechanisms, we added your recommended paper (Plassmeyers et al., 2023) L224-225. Also, we added your recommended paper regarding the DHA study (Chizhikov et al., 2020) in the High-fat diet paragraph (L315-316).

In summary, we appreciate suggestions of both reviewers and hope it has significantly strengthened the presentation of our review.

Round 2

Reviewer 1 Report

Comments and Suggestions for Authors

Fine revision of the manuscript.

Reviewer 2 Report

Comments and Suggestions for Authors

I recommend to accept this manuscript in its present form.